# Rectal Epithelial Stem Cell Kinetics in Acute Radiation Proctitis

**DOI:** 10.3390/ijms252011252

**Published:** 2024-10-19

**Authors:** Sharmila Ghosh, Akinori Morita, Yuichi Nishiyama, Masahiro Sakaue, Ken Fujiwara, Daiki Morita, Yuichiro Sonoyama, Yuichi Higashi, Megumi Sasatani

**Affiliations:** 1Graduate School of Biomedical Sciences, Tokushima University, Tokushima 770-8503, Japan; ghoshsharmila0427@gmail.com (S.G.); y-nishi@tokushima-u.ac.jp (Y.N.); sakamasa135@gmail.com (M.S.); ken.217015@gmail.com (K.F.); dmori5943@gmail.com (D.M.); sono.0513yuu@gmail.com (Y.S.); y.higashi.radiation@gmail.com (Y.H.); 2Research Institute for Radiation Biology and Medicine, Hiroshima University, Hiroshima 754-8553, Japan; mtoyosh@hiroshima-u.ac.jp

**Keywords:** mouse model, radiation proctitis, *Lgr5*-positive stem cell, crypt base regeneration, caudal half-body irradiation

## Abstract

The intestinal tract is a typical radiosensitive tissue, and radiation rectal injury is a severe side effect that limits the prescribed dose in radiotherapy of the abdominal and pelvic region. Understanding the post-irradiation kinetics of *Lgr5*-positive stem cells is crucial in comprehending this adverse process. In this study, we utilized *Lgr5*-*EGFP* knock-in mice expressing EGFP and LGR5 antibody fluorescence staining of wild-type mice. At the state of radiation injury, the qPCR analysis showed a significant decrease in the expression level of *Lgr5* in the rectal epithelial tissue. The dose-response relationship analysis showed that at low to moderate doses up to 10 gray (Gy), *Lgr5*-clustered populations were observed at the base of the crypt, whereas at sublethal doses (20 Gy and 29 Gy), the cells exhibited a dot-like scatter pattern, termed *Lgr5*-dotted populations. During recovery, 30 days post-irradiation, *Lgr5*-clustered populations gradually re-emerged while *Lgr5*-dotted populations declined, implying that some of the *Lgr5*-dotted stem cell populations re-clustered, aiding regenerations. Based on statistical analysis of the dose-response relationship using wild-type mice, the threshold dose for destroying these stem cell structures is 18 Gy. These findings may help set doses in mouse abdominal irradiation experiments for radiation intestinal injury and for understanding the histological process of injury development.

## 1. Introduction

The acceptable levels of radiation exposure for nearby healthy cells circumscribe the administration of radiation therapy to pelvic malignancies putting organs at risk, i.e., the colon, rectum and bladder [1]. Most abdominal and pelvic radiotherapy regimens commonly affect the colon and rectum due to their anatomical location [2]. Rectal toxicity often becomes the limiting factor in pelvic radiation therapy due to dose considerations [3]. Up to 75% of patients undergoing pelvic radiotherapy may experience rectal symptoms [4,5]. High radiation doses can result in radiation-induced gastrointestinal syndrome (RIGS), leading to the death of the intestinal cells [6,7]. Radiation proctitis, a common complication affecting nearly half of the patients with pelvic malignancies undergoing radiation therapy, manifests as diarrhea, intestinal bleeding, dehydration, and sepsis [8,9]. The recent establishment of non-invasive management for rectal cancer underscores the increasing importance of understanding the radiosensitivity of the rectal epithelium and its recovery process. This knowledge is crucial for optimizing treatment strategies and improving patient outcomes in radiotherapy [10,11].

The rapidly dividing Intestinal epithelial cells are replaced every 4–5 days and undergo apoptosis at the villi’s tips [12,13,14]. The structural integrity is maintained by the intestinal stem cells (ISCs) at the base of the crypt [15,16]. Depletion of ISCs is the primary cause of the damage caused by irradiation, as they give rise to diverse cell types that comprise the intestinal epithelium. Thus, ISCs are the critical regulators for regaining intestinal homeostasis after radiation damage [14,17].

One of the widely studied intestinal stem cell markers, leucine-rich repeat-containing G protein-coupled receptor 5 (*Lgr5*), also known as Grp49, was first identified in 2007 [18]. Radiation explicitly targets cells that proliferate faster, such as those found in the intestinal crypt. *Lgr5*-positive cells play a crucial role in the regeneration from radiation-induced damage in the small bowel [19,20,21]. Many studies have also investigated *Lgr5* stem cells as a marker in colorectal studies [22,23,24,25]. However, there is little understanding of the radiation-induced damage to rectal *Lgr5*-positive stem cells and their role in repair. Furthermore, the involvement of other stem cells in the regenerative response to radiation injury has yet to be extensively studied.

This study aimed to examine the kinetics of *Lgr5* rectal stem cells during the injury and recovery phase and their participation in crypt regeneration. We employed two methodologies: enhanced green fluorescent protein (EGFP) observation using *Lgr5*-*EGFP* mice and LGR5 observation by immunofluorescence using wild-type mice. *Lgr5*-*EGFP* mice are genetically engineered to express an EGFP under the control of the *Lgr5* promoter in *Lgr5*-expressing intestinal cells, aiding in the visualization and study of intestinal stem cells [18]. We used caudal half-body irradiation (CHBI) under moderate to sublethal irradiation doses. Few studies have used rectal abdominal irradiation above 25 gray (Gy) [26,27]. Other studies have reported that a whole abdominal dose lower than 17.5 Gy can show intestinal stem cell regeneration, whereas doses above 20 Gy inhibit ISC regeneration and repair following irradiation [28,29]. However, our findings indicate that *Lgr5*-positive rectal stem cells were able to recover 30 days after exposure to 20 Gy and 29 Gy CHBI. *Lgr5*-positive rectal stem cells’ ability to self-regenerate post high irradiation doses offers a new therapeutic insight for radiation proctitis. Determining the maximal dose that is well tolerated in the mouse model is crucial. High-dose radiation mouse models are vital for testing the efficacy of potential radioprotectors or mitigators for treatments. Our study will help to determine the radiation dose required for structural disruption of rectal epithelial stem cells. 

## 2. Results

### 2.1. Immunofluorescence Assay of Rectal Epithelial Stem Cells

We initially tested the rectal tissue for LGR5 stem cell expression using immunofluorescence alongside representative stem cell markers commonly expressed in the small intestine and colon, such as olfactomedin 4 (OLFM4) [30] and achaete-scute complex homolog 2 (ASCL2) [31,32]. In the basal crypt of jejunal tissue, which was utilized as a positive control, both LGR5 and ASCL2 demonstrated positive staining, leading to an overlap of green LGR5 and red ASCL2 signals, which appeared as yellow (Figure 1A). Furthermore, OLFM4 staining was also observed in the basal crypt of the jejunum; however, its co-localization with LGR5-positive cells was only partial (Figure 1B). In contrast, rectal tissue exhibited positive LGR5 expression (Figure 1), but no detectable signal for ASCL2 or OLFM4 was observed. This analysis determined that the only suitable marker molecule for detecting rectal stem cells is LGR5. 

### 2.2. Intestinal Damage by CHBI: Damage Analysis by Gut Segment

The CHBI setup was designed to precisely target the lower half of the body while shielding the cranial part, with a 135 mm lead diameter specifically chosen to target the rectal tissue (Appendix A). A single dose of 29 Gy was irradiated to induce acute radiation proctitis, leading to gradual but substantial damage to intestinal cells and tissue (Appendix A). Histopathological examination of sections along the intestinal tract after 7 days revealed that the damage is relatively low and consistent in the unirradiated group (Appendix A). In contrast, according to the relationship between the anatomical location of each bowel segment and the irradiated field, substantial damage across sections of the intestinal tract revealed the lowest damage in the duodenum, progressively increasing towards the colorectum (Figure 2A). The irradiated colorectal tissue exhibited 45% damage, showing a statistically significant difference (Figure 2B). The ileum sustained less damage, likely due to its distal location from the jejunum and reduced exposure to the highest concentration of irradiated particles [33]. After confirming that the highest damage had occurred in the colorectum, the expression of *Lgr5* in the colon and rectum was examined by quantitative PCR (qPCR) from surgically resected tissue. *Lgr5* expression levels differed significantly between dead and alive mice, so 25 Gy was selected as the dose at which most mice would survive. A significant decrease in *Lgr5* expression in the rectum was observed 72 h post-irradiation (Figure 2C,D). The percentage of live cells was estimated at 25%. These data sets support that the damage has occurred primarily in the rectum and confirm that these conditions are optimal for maximizing rectal damage.

### 2.3. Dose Response of Rectal Epithelial Lgr5-EGFP-Positive Cells 9 Days After Irradiation

After confirming the *Lgr5* expression was significantly reduced in the rectum, the effects of different radiation doses on the *Lgr5-EGFP*-positive cell populations in mice were investigated. The *Lgr5-EGFP* knock-in mice were divided into groups: unirradiated and irradiated groups of 10 Gy (moderate dose), 20 Gy and 29 Gy (sublethal doses) to give an insight into the behavior of *Lgr5*-positive stem cells under varying irradiation doses. After 9 days, rectal tissue was obtained, and subsequent analysis revealed distinct patterns in the distribution of *Lgr5-EGFP*-positive cells. Two types of EGFP populations were observed: EGFP-clustered populations were observed mainly under unirradiated, and moderate dose conditions (Figure 3A upper panel) and EGFP-dotted populations were observed mainly under sublethal doses (Figure 3A lower panel). The basal value of the clustered populations was 3.8 per circumference in the unirradiated group, which increased in the 10 Gy irradiated group. However, a significant reduction was observed at higher doses, with the value decreasing to 0.30 per circumference at 20 Gy and a further reduction to 0.10 per circumference at 29 Gy. Conversely, the opposite trend was seen with the dotted cell populations, which had the highest value of 250 and 200 per circumference in 29 Gy and 20 Gy irradiated mice, respectively (Figure 3B). These data show an inversely proportional relationship between the two EGFP populations, suggesting that the clustered cell populations were structurally disrupted and scattered after irradiation. To further investigate the correlation between EGFP-clustered populations and crypt integrity, the number of intact crypts was counted in unirradiated and irradiated samples (Figure 3C). As the irradiation doses increased, the crypt microcolony assay [34,35] showed a dose-dependent reduction in the regenerated crypts. A significant reduction was observed at higher doses, with the crypt number falling to 22 crypts per circumference at 20 Gy and further lowering to 6.0 per circumference at 29 Gy. This finding suggests a direct relationship, indicating that clustered *Lgr5*-positive stem cells are responsible for maintaining the structural integrity of crypts.

### 2.4. Effects of High-Dose Radiation on EGFP Kinetics in Lgr5-EGFP Mice: A 60-Day Study

After confirming the existence of two types of EGFP populations, long-term observations were required to understand the dynamics of dotted-cell populations and the role of clustered populations in the recovery phase. For long-term observations, the first method we utilized was *Lgr5-EGFP* knock-in mice to study the kinetics of the EGFP cell population under high-irradiation doses (20 Gy and 29 Gy). During the recovery phase, EGFP-clustered populations showed a gradual reappearance in irradiated mice. In the 20 Gy irradiated group, the number of EGFP-clustered populations recovered to 1.2 per circumference (Figure 4A), and in the 29 Gy irradiated group, it recovered to 0.69 per circumference (Figure 4B), compared to the basal value of 3.5 (Appendix A) on day 0, indicating approximately 32% and 20% recovery, by the end of 60 days. Correspondingly, the number of EGFP-dotted populations gradually decreased, with 120 and 160 per circumference, representing a 58% and 64% reduction compared to day 9 in the 20 Gy and 29 Gy irradiated groups, respectively. The EGFP-clustered populations slowly started to reappear 30 days post-irradiation. Some EGFP-clustered populations survived post-irradiation in the 20 Gy irradiated mice, facilitating a faster regeneration process (Appendix A). In parallel, the number of regenerated crypts was 64 per circumference by 60 days, showing a nearly 69% recovery in the number of crypts (Figure 4C). However, the reappearance of the EGFP-clustered populations was delayed in 29 Gy irradiated mice 45 days post-irradiation (Appendix A), regenerating with 29 crypts per circumference, and a 31% recovery by day 60 (Figure 4C). We can consider cryptic regeneration and clustering to be the same event. This finding suggests that the EGFP-clustered populations can regenerate back after high irradiation.

### 2.5. Time Course of Stem Cell Kinetics and Determination of the Threshold Dose

The second method for long-term observation involved using ICR wild-type mice and studying LGR5 kinetics by immunofluorescence under high irradiation doses (20 Gy and 29 Gy). Sixty days post-irradiation, the number of LGR5-clustered populations was 2.0 and 1.4 per circumference from a basal value of 5.0 (Appendix A), indicating 40% and 28% recovery in 20 Gy and 29 Gy, respectively (Figure 5B,C). In the second group, LGR5-dotted populations gradually decreased per circumference to 84 and 150, decreasing by 52% and 57% compared to day 10 in the 20 Gy and 29 Gy irradiated groups (Figure 5B,C). The LGR5-clustered populations began to reappear 30 days post-irradiation in both irradiated groups along with some LGR5-clustered populations surviving post-irradiation in the 20 Gy irradiated group (Figure 5A). Simultaneously, crypt recovery was observed at 70% and 40% in 20 Gy and 29 Gy CHBI mice (Figure 5D).

Both the methods used to analyze the *Lgr5* stem cell kinetics demonstrated a strong correlation between the recovery of clustered populations (Appendix A). Doses above 20 Gy were effective in damaging the *Lgr5* stem cell. Therefore, the dose range was limited to 10 Gy to 20 Gy to identify the exact dose below which the damage in stem cell structure does not occur. To analyze the effects of irradiation and the subsequent recovery, rectal samples were collected at the 30-day mark (Appendix A). This interval was selected to capture tissue conditions before complete recovery, thereby emphasizing the extent of damage and the ability to recover from different doses within the selected range. The number of LGR5-clustered populations decreased with increasing doses, with a statistically significant difference at 18 Gy. This dose was identified as the threshold dose for rectal stem cell structural destruction (Figure 5E) [28].

## 3. Discussion

Radiation proctitis is highly observed in rectal tissues as a side effect of irradiation, which causes epithelial damage, leading to rectum inflammation [36,37,38]. We designed our study to focus irradiation damage specifically on the rectum and maximize it by selecting sublethal doses. It was imperative to use doses that could efficiently damage the stem cell structure to understand its role in the state of recovery. The selection of high doses made it possible to uncover the different kinetics of *Lgr5* stem cells. Two distinct cell populations were identified: the *Lgr5*-positive cluster cell populations in the non-irradiated state and the *Lgr5*-positive dotted cell populations during the radiation injury process, which are newly discovered cell structures in the rectum.

Our study demonstrated that *Lgr5*-clustered populations, typically observed as around five populations per circumference within the rectal epithelium, seem to play a vital role in preserving the integrity of the epithelial barrier (Appendix A). These populations are particularly sensitive to high-dose irradiation (CHBI) and are rapidly reduced when exposed to irradiation levels above a specific threshold, resulting in the spread of cells into *Lgr5*-dotted populations. Despite this initial depletion, the *Lgr5*-clustered populations show a marked increase during the recovery phase and the reappearance of intact crypts, particularly after 30 days, highlighting their critical role in the regeneration process. This suggests that *Lgr5*-clustered populations not only help maintain epithelial stability under normal conditions but are also key players in the tissue’s recovery and repair following significant injury. Our study indicated a potential difference in the response of *Lgr5* marker stem cells under different doses of radiation. As a response to injury, there appeared to be a significant reduction in the expression of *Lgr5* stem cells after 72 h. At the 9 days mark, the dose of 10 Gy showed no change in pathological condition in the rectal tissue [39]. EGFP-clustered populations seemed more prevalent in the unirradiated group of mice and those subjected to 10 Gy irradiation. Mice irradiated with 10 Gy exhibited slightly higher clustered populations compared to unirradiated mice, which may be related to the phenomenon of the increased recruitment of cells into the clonogenic compartment under moderate doses, as previously reported [40]. This observation suggests potential protective or regenerative responses under low doses of radiation in the rectum. At higher doses, the irradiated mice show a reduction in EGFP-clustered populations at the crypt base, with the EGFP populations possibly spreading into a dotted pattern. This pattern appeared more prevalent in the groups exposed to 20 Gy and 29 Gy. According to previous studies, it has been suggested that stem cells in the small intestine are the primary site of apoptosis, and in the colon, most of the TUNEL^+^ cells were positive for *Lgr5-EGFP* [41,42].

The state of radiation injury demonstrated the importance of *Lgr5* survival in rectum crypt regeneration. However, some questions remained: Are the dotted cells seen after high doses of irradiation a consistent structure or will they disappear in the recovery phase? Can the lost clustered stem cell population recover after higher doses of irradiation? To understand this, we conducted a long-term recovery state analysis; to increase the efficiency and robustness of the analysis, we used two strains of mice: *Lgr5-EGFP* knock-in mice and ICR wild-type mice. In the 29 Gy irradiated group, it appeared that there was a catastrophic loss of *Lgr5*-clustered populations and crypt sterilization after irradiation [43,44]. Recovery began slowly, with a new population of *Lgr5*-clustered populations reappearing 30 days post-irradiation, with few crypt structures seen to regenerate. The 20 Gy irradiated group appeared to experience less severe loss of *Lgr5*-clustered populations and crypt structure. Some *Lgr5*-clustered populations seemingly survived post-irradiation, potentially aiding in the repopulation of the stem cell population and faster recovery of the crypt structure. The *Lgr5*-dotted populations were seen to gradually decline over time but were not completely lost. By the end of 60 days of observation, both the irradiated groups appeared to have recovered reasonably well compared to the conditions seen after irradiation, with an increase in the numbers of *Lgr5*-clustered populations and a decline in *Lgr5*-dotted populations, implying that some of the dotted populations re-clustered and regenerated as stem cells (Figure 6). Our study reveals that *Lgr5* stem cells can regenerate in the rectum even after being completely lost after high-dose irradiation. However, the exact mechanism behind this regeneration remains unclear. To clarify the underlying mechanism of this process, we are currently performing an in-depth RNA sequencing analysis to examine the gene expression profiles associated with regeneration. Further research should focus on identifying factors that promote re-clustering of dotted cells.

Alternatively, we observed faster recovery of ICR wild-type mice compared to *Lgr5-EGFP* knock-in mice. We suspect that this is due to potential haploinsufficiency induced by the loss of one *Lgr5* allele. The reporter cassette is prone to be silenced in over two-thirds of all the crypts, resulting in mosaic expression in *Lgr5-EGFP* knock-in mice [16,45]. The threshold dose for the damage of the rectal stem cell structure was determined to be 18 Gy; this might be helpful in studies to evaluate the effectiveness of potential radioprotective agents in mitigating or preventing damage at this critical dose. The reason for stem cell regeneration after irradiation is still uncertain. We speculate that *Lgr5*-dotted populations play an important role in the recovery phase. The dotted cell populations may be a mixed population of cells where most cells undergo apoptosis/senescence, while some undergo DNA repair. These repaired cells can possibly be the cells that cluster back to the crypt base and help in the repair and regeneration of the crypt. Some studies have shown an increase in 53BP1-GFP foci after irradiation [46,47], with 17% migrating and merging to form clustered DSB repair centers post X-ray irradiation [48,49,50]. Some previous reports claimed that the Wnt signal pathway was activated again to resume the *Lgr5* expression at the crypt base or that regeneration can be seen due to the presence of reserve stem cell polls [51,52].

The findings from our study suggest that *Lgr5* stem cells in the rectum possess the ability to regenerate, even after being destroyed by high levels of irradiation. However, the exact mechanism behind this regeneration remains unclear. Further research is needed to identify the factors responsible for this recovery.

## 4. Materials and Methods

### 4.1. Mice

Heterozygous Lgr5-EGFP-IRES-creERT2 “knock-in” allele mice were imported from the Jackson Laboratory in the United States. When purchased, it was a C57BL/6J strain, but by backcrossing with the ICR strain for at least 4 generations, the genetic background was replaced with that of ICR mice as much as possible (the latest was the 12th generation of backcrossing). The *Lgr5-EGFP* mice pups were genotyped, and the female mice were used at 8 weeks. The wild-type female ICR mice (7 weeks of age) were purchased from sankyo labo service corporation (Shizuoka, Japan). They were allowed to acclimatize in a laboratory environment set at 22 °C with a light-dark cycle, receiving food and water ad libitum. The animals’ health was observed once every day throughout the study, and death was confirmed by examining the carcasses. The humane endpoint was determined with a 20% decrease in body weight over 7 days. A total of approximately 88 *Lgr5-EGFP* mice and 130 ICR wild-type mice were used in this study. The specific number of mice per group size (*n*) ranged from a minimum of 3 to a maximum of 10 except in Figure 4B,C due to mouse mortality caused by high doses of irradiation. We employed block randomization to assign an equal number of mice from each irradiation group to different cages and used a single-blinded approach with masked data during analysis to minimize bias. The total duration of the experiment was between May 2022 to August 2024. The animal experimental committee at Tokushima University approved the experiment (protocol code T2022-96).

### 4.2. Induction of Acute Radiation Proctitis 

After acclimatizing the mice for one week, the mice were anesthetized by intraperitoneal injection with a mixture of midazolam (4 mg/kg), medetomidine (0.3 mg/kg) and butorphanol (5 mg/kg) 20 min before irradiation [53]. Anesthetic reagent was used to accurately irradiate the caudal half-body and to reduce immobilization stress during the irradiation treatment. A single large local dose of 20 Gy and 29 Gy was established to induce acute radiation proctitis in mice. CHBI was performed where the cranial half-body was shielded with the lead of ϕ135 mm × t18 mm to avoid bone marrow death, and the caudal half of the body was irradiated using an X-ray generator (MBR-1520R-3, Hitachi, Tokyo, Japan) with a dose rate of approximately 1.0 Gy/min. Dosimetry was carried out with a 0.3 cc N31003 ionization chamber (PTW Freiburg, Freiburg, Germany).

### 4.3. Morphology of Irradiated Intestine

To study the damage caused by radiation, wild-type female mice (8 weeks old) were irradiated with 29 Gy CHBI and were sacrificed after 168 h (7 days). The entire length of the intestine was removed. It was divided into four parts: duodenum, jejunum, ileum and colorectum. After collecting the sample, we flushed the intestine contents with phosphate-buffered saline (PBS). We opened the specimens longitudinally, rolled them from the proximal side and tied them with a thread fixed in 10% formalin and embedded in paraffin [54]. After 5 µm thick sections were obtained, they were stained with hematoxylin and eosin (HE). The epithelium damage after irradiation was estimated using BZ-9000 fluorescence microscopy (Keyence, Osaka, Japan). Damage percentage was determined by the length of damage area/total length of the sections × 100.

### 4.4. Visualization of LGR5/EGFP Population

The mouse rectum was fixed in 4% paraformaldehyde (PFA) for 30 min at 4 °C, followed by immersion in 10% and 20% sucrose solution for 1 h each. The tissue was then embedded in an optimal cutting temperature compound cryostat sectioning medium and sectioned at 8 µm. After tissue preparation, the sections were washed in PBS for 15 min (2 times) and mounted with VECTASHILED mounting medium with DAPI (Vector Laboratories, Burlingame, CA, USA). The population of LGR5/EGFP-positive cells was observed using an A1R confocal microscope (Nikon, Tokyo, Japan) and a BZ-9000 fluorescence microscope (Keyence). The population of cells at the base of the crypt was counted manually and named clustered populations with a population size of 4~5 cells to provide the signal intensity for each group recorded as a single count. The *Lgr5*-clustered populations were counted ensuring an intact crypt with a distinct signal originating from the crypt base. The other population of cells was named *Lgr5*-dotted populations, characterized by their distinct circular shape which spread into dotted patterns after irradiation, and were counted using Image J version 1.53 (free available software, https://imagej.net/ij/ (accessed on 1 August 2024)) per circumference with a population size of 1~3 cells per group. The intact crypt structure was manually counted per circumference which was well connected to the mucosa and the epithelium structure was not highly damaged.

### 4.5. Quantitative PCR (qPCR)

qPCR analyses were performed according to the previous report [55]. Briefly, 72 h after 25 Gy irradiation, the mouse colorectum analysis was conducted. The mouse colon and rectum were surgically dissected with careful removal of the mesentery, washed inside and outside several times with ice-cold PBS, and longitudinally sliced open. The epithelium was expanded on a flat stainless-steel plate with its surface up and then the plate was put on an aluminum cube which was kept in a freezer at −80 degrees Celsius before use. Epithelial tissue was then carefully collected by immediately soaking and freezing rectal mucosa with TRIzol reagent (Invitrogen, Carlsbad, CA, USA) on the cooled plate and scraping one-third from the epithelial surface with a surgical blade. Total RNA was then extracted from the TRIzol solution according to the manufacturer’s instructions. cDNA synthesis was performed using ReverTra Ace^®^ qPCR RT Master Mix (TOYOBO, Osaka, Japan) and incubated using PCR Thermal Cycler Dice (TaKaRa, Shiga, Japan). Real-time PCR was conducted on the StepOnePlus system (Applied Biosystems, Waltham, MA, USA) using THUNDERBIRD^®^ SYBR^®^ qPCR Mix (TOYOBO). Gene expression levels were compared using the ΔΔCt method. The primer sequences used for the analyses were as follows: 

Lgr5

Forward: cctcactcggtgcagtgct

Reverse: cagccagctaccaaataggtg

### 4.6. Immunofluorescence

Frozen sections (8 µm) from ICR wild-type mice were washed twice with PBS, followed by a 1 h treatment with blocking buffer (1% bovine serum albumin (BSA) in PBS with 0.1% TritonX-100 (PBS-T)). After aspirating the blocking buffer, the primary antibodies, Lgr5/GPR49 (clone MAB8240, R&D systems, Minneapolis, MN, USA, 1:400), Olfm4 (clone 39141, Cell Signaling Technology, Danvers, MA, USA, 1:1000) and Ascl2 (orb155740, Biorbyt, Cambridge, UK, 1:1000) diluted in 1% BSA in PBS-T, were added to the section and incubated for 3 days at 4 °C in a humidified chamber. Sections were then washed three times with PBS for 15 min each. We added fluorescent label conjugate secondary antibodies, Goat anti-Rabbit IgG (Alexa Fluor 546, A-11010, Thermo Fisher Scientific, Waltham, MA, USA, 1:1000), Goat anti-Rat IgG H&L (Alexa Fluor 488, abcam150157, Abcam, Cambridge, UK, 1:1000) and Goat anti-Rat IgG H&L (Alexa Fluor 546, A-11081, Thermo Fisher Scientific, 1:1000) diluted with 1% BSA in PBS-T, and incubated the sections overnight at room temperature in the dark. We washed the sections three times with PBS for 15 min and mounted them with VECTAHIELD mounting medium with DAPI (Vector Laboratories). All the stem cell-positive signals were observed by using a BZ-9000 fluorescence microscope (Keyence).

### 4.7. Statistics

All data are presented as the mean ± standard deviation (SD). Statistical analyses were performed with at least three mice per group (*n* ≥ 3), except in Figure 4B,C, where a sample size of *n* = 2 precluded statistical analysis due to mortality associated with high-dose irradiation. Student’s *t*-test was used to determine *Lgr5* expression in colorectal mucosa. Dunnett test was used to determine the most damaged intestinal section, the dose- and time-dependent changes in clustered and dotted populations, and the time-dependent recovery of the crypt and threshold dose. Multiple regression analysis was used to find the relation between the clustered population and the number of crypts, with the independent variable as the dose and the dependent variable as the crypt count and clustered population. Linear regression analysis was used to determine the relationship between the two methods for stem cell analysis, and chi-square tests for survival percentage were used for statistical analysis. A *p*-value of <0.05 was considered to indicate statistical significance.

## Figures and Tables

**Figure 1 ijms-25-11252-f001:**
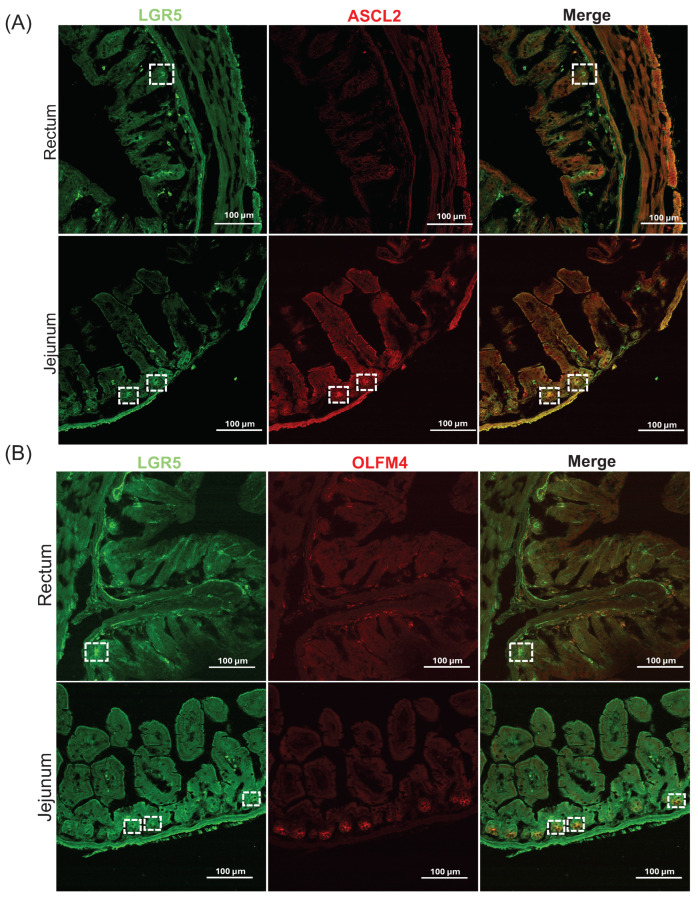
LGR5 is a suitable marker for detecting stem cells in the rectal crypts. Immunofluorescence staining of the rectum and the jejunum of ICR wild-type mice for (**A**) co-staining of LGR5 (green) and ASCL2 (red) and (**B**) co-staining of LGR5 (green) and OLFM4 (red). The upper panel is for the rectum (test tissue) and the lower is for the jejunum (positive control). Positive signal marked in white boxes. Magnification 20×. Scale bar: 100 μm. Achaete–scute complex homolog 2 (ASCL2); leucine-rich repeat-containing G protein-coupled receptor 5 (LGR5); olfactomedin 4 (OLFM4).

**Figure 2 ijms-25-11252-f002:**
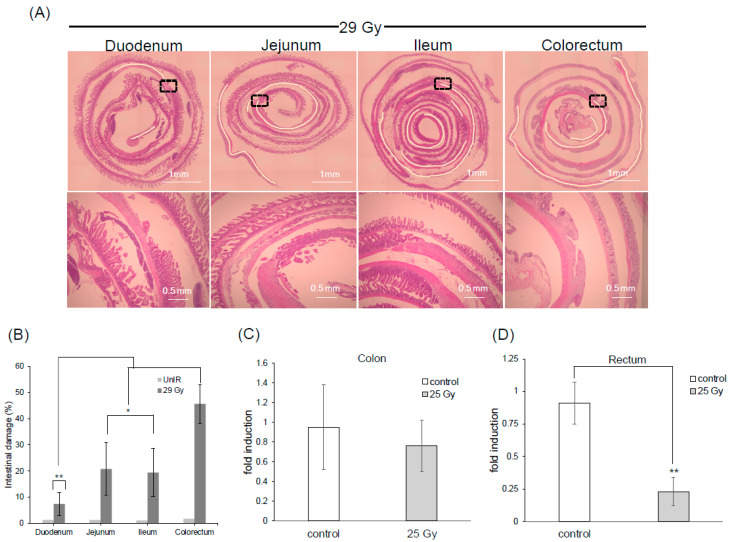
Rectum is highly damaged by CHBI. (**A**) Representative images of ICR wild-type mice with hematoxylin and eosin staining of the entire intestinal tract of the 29 Gy CHBI group were obtained after 7 days (168 h). The images depict various segments of the intestine. The damaged areas are traced and highlighted in yellow in the upper panel. The border between damaged and healthy regions is magnified in the lower panel. (**B**) Intestinal damage percentage. qPCR analysis of *Lgr5*-positive stem cell. (**C**) Colon. (**D**) Rectum. Magnification 4×. Scale bar: 1 mm (upper panel), 0.5 mm (lower panel). Data represent the mean ± standard deviation. Asterisks indicate a significant difference (* *p* < 0.05; ** *p* < 0.01) between the colorectum vs. other sections and control vs. irradiated mice (25 Gy). Caudal half-body irradiation (CHBI); gray (Gy); quantitative PCR (qPCR); unirradiated (UnIR).

**Figure 3 ijms-25-11252-f003:**
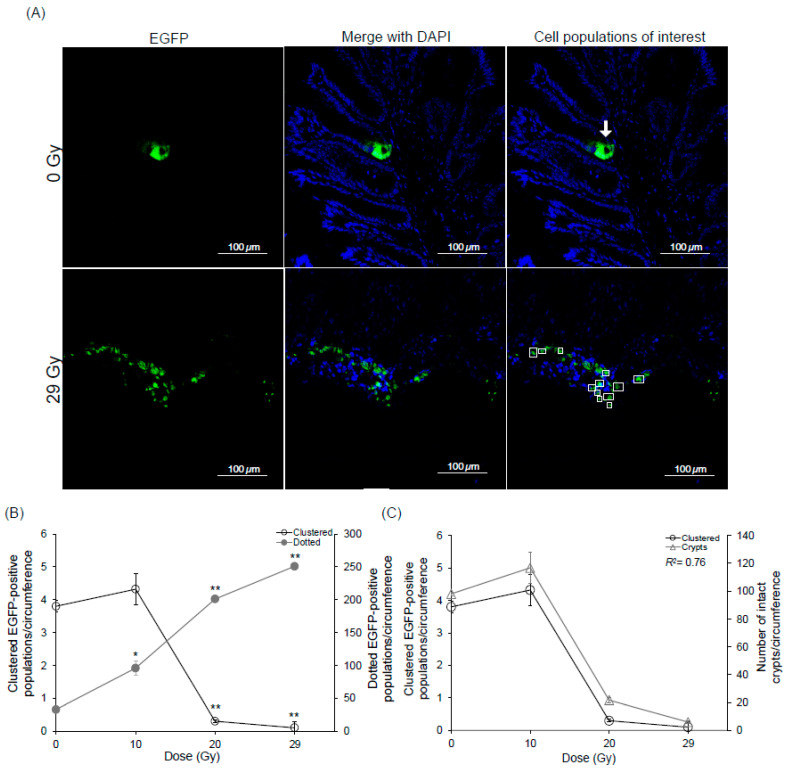
Different doses display different EGFP populations. (**A**) Representative images of EGFP-clustered populations at the base of the rectal crypt and EGFP-dotted populations spread after irradiation. Arrows indicate representative clustered populations. Small dashed squares indicate representative dotted populations. (**B**) Graphical presentation of the response of EGFP population in *Lgr5*-*EGFP* mice (216 h) in unirradiated and CHBI conditions at 10 Gy, 20 Gy and 29 Gy after 9 days. (**C**) Correlation between clustered EGFP-positive cells versus the number of crypts unirradiated, 10 Gy, 20 Gy and 29 Gy (*R*^2^ = 0.76). Magnification 40×. Scale bar: 100 μm. Data represent the mean ± standard deviation. Asterisks indicate a significant difference (* *p* < 0.05; ** *p* < 0.01) between unirradiated and irradiated groups (10 Gy, 20 Gy and 29 Gy). Caudal half-body irradiation (CHBI); 4′,6-diamidino-2-phenylindole (DAPI); enhanced green fluorescent protein (EGFP); gray (Gy); leucine-rich repeat-containing G protein-coupled receptor 5 (*Lgr5*).

**Figure 4 ijms-25-11252-f004:**
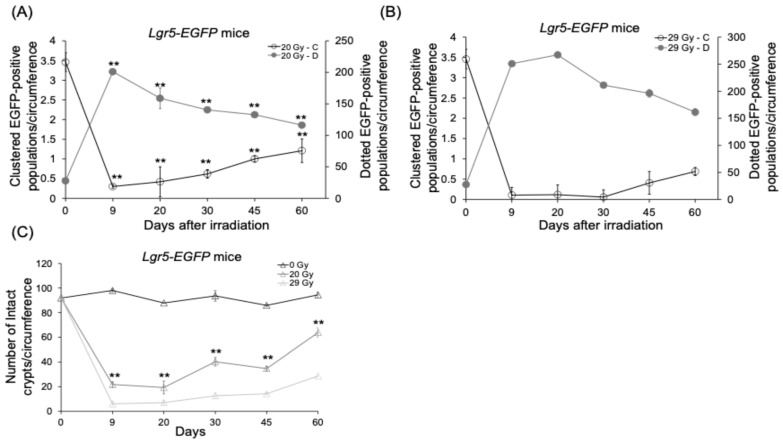
High-dose irradiation promotes regeneration in EGFP populations with crypts. Recovery of stem cell kinetics in 60 days in *Lgr5-EGFP* mice. (**A**) 20 Gy CHBI. (**B**) 29 Gy CHBI. (**C**) Crypt microcolony survival assay. At certain time points, the mice number was reduced to *n* = 2 due to high-dose mortality. Data represent the mean ± standard deviation. Asterisks indicate a significant difference (** *p* < 0.01) between 0 days and the subsequent time points (9 days, 20 days, 30 days, 45 days and 60 days). Caudal half-body irradiation (CHBI); enhanced green fluorescent protein (EGFP); gray (Gy); leucine-rich repeat-containing G protein-coupled receptor 5 (*Lgr5*).

**Figure 5 ijms-25-11252-f005:**
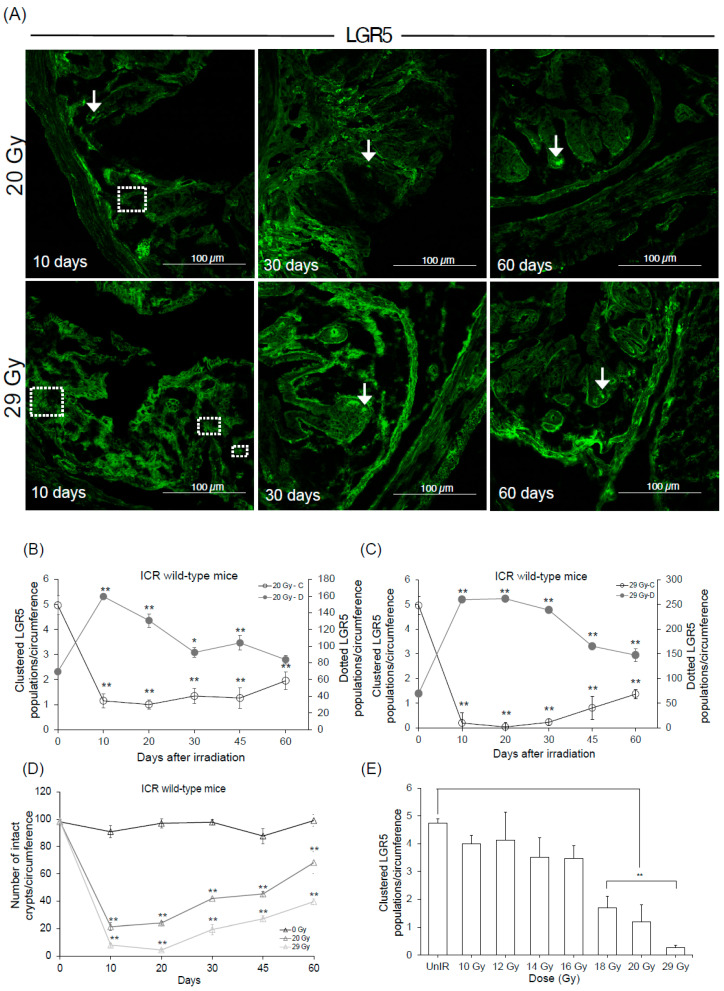
ICR wild-type mice show faster recovery compared to *Lgr5*-*EGFP* knock-in mice. (**A**) Representative images of reappearances of the LGR5 population post-irradiation (10 days to 60 days). Arrows indicate representative clustered populations. Dashed squares indicate representative dotted populations. Days versus LGR5-positive stem cell relation in (**B**) 20 Gy CHBI, (**C**) 29 Gy CHBI. (**D**) Crypt microcolony survival assay. (**E**) Determining threshold dose after 30 days. Data represent the mean ± standard deviation. Asterisks indicate a significant difference (* *p* < 0.05, ** *p* < 0.01) between 0 days and the subsequent time points (9 days, 20 days, 30 days, 45 days and 60 days) and between the unirradiated group and other irradiated groups. Magnification 20×. Scale bar: 100 μm. Caudal half-body irradiation (CHBI); enhanced green fluorescent protein (EGFP); gray (Gy); leucine-rich repeat-containing G protein-coupled receptor 5 (LGR5).

**Figure 6 ijms-25-11252-f006:**
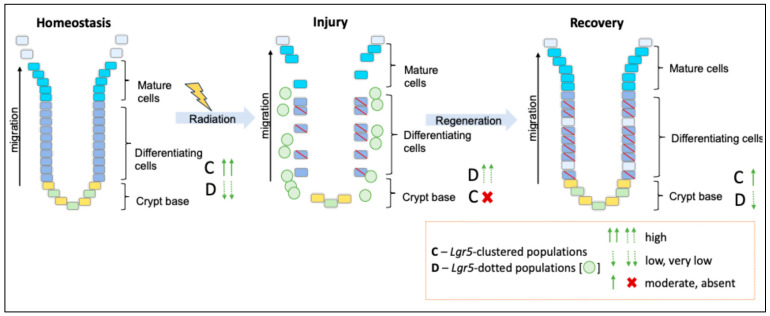
A proposed model to show the spread of *Lgr5*-positive stem cells in response to higher doses of irradiation and cluster to its original state during the state of recovery. The green circles in the injury phase represent the *Lgr5*-dotted population. Leucine-rich repeat-containing G protein-coupled receptor 5 (*Lgr5*).

## Data Availability

The data that support the findings of this study are available from the corresponding author, Akinori Morita, upon reasonable request.

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
