# Peer review of "Rectal Epithelial Stem Cell Kinetics in Acute Radiation Proctitis"

_ijms, 2024, doi:10.3390/ijms252011252_

Round 1

Reviewer 1 Report

Comments and Suggestions for Authors

In the manuscript entitled “Rectal Epithelial Stem Cell Kinetics in Acute Radiation Proctitis”, author has tried toshow the kinetics of the rectal stem cell regeneration. 

Author has applied lethal doses of 20 Gy and 29 Gy CHBI (Caudal Half-Body Irradiation) to develop proctitis. The manuscript suffers from following flaws -

1.    The mechanism behind the regeneration of rectal stem cells after 60 days post IR is not explained and proved. Author should explain and demonstrate the mechanism behind this hypothesis. 

2.    Author has shown two different types of LGR5 population: Clustered and single EGFP+cells.  From the staining (Figure 3-B) it is not clear where these single cells are located. Author should cross validate these single cells as LGR5+ cells using another marker as well.  It is already reported that there are regions in crypt where LGR5 cell population is high, medium and low. Author should use some other method to show the location of these single cells. 

3.    In the method author mentioned that “The epithelial mucosa was carefully collected by soaking and freezing with TRIzol reagent (Invitrogen) and by scraping with a surgical blade”. Author should isolate the epithelial cells intead of mucosa, as mucosa contains all other stromal cells. So, all the qPCR analysis should be done on the Intestinal epithelial cells. 

4.    The IF image quality is very poor (Fig1, 3, 5)

5.    The experiments should be repeated using the clinical dose of fractionated radiation instead of single radiation dose to create the scenario of clinical proctitis as it develops in patients following radiation therapy.    

Comments on the Quality of English Language

Average

Author Response

  1. The mechanism behind the regeneration of rectal stem cells after 60 days post IR is not explained and proved. Author should explain and demonstrate the mechanism behind this hypothesis.

Thank you for your valuable comment.

In response to this, we have revised the discussion section of our manuscript to provide a more comprehensive explanation of the potential mechanisms underlying stem cell regeneration in this context. The changes are added in section 3 (lines 261-267) Specifically, we are currently conducting an in-depth RNA sequencing (RNA-seq) analysis to investigate the gene expression profiles associated with this regenerative process. Our focus is on identifying gene groups that show a rapid decline during the acute injury phase and those that demonstrate a significant upregulation during the recovery phase.

One of the significant challenges we have encountered in this work is the extraction of high-quality RNA from the rectum, a digestive organ with a high concentration of hydrolyzing enzymes that can degrade RNA. Maintaining RNA purity and integrity under these conditions has been particularly difficult. However, we have recently established a method that overcomes these challenges, enabling us to obtain RNA samples of sufficient quality for reliable downstream analysis.

Our next research plan involves analyzing the candidate genes identified through this experiment, to elucidate the details of the regeneration mechanism.

2. Author has shown two different types of LGR5 population: Clustered and single EGFP+cells. From the staining (Figure 3-B) it is not clear where these single cells are located. Author should cross-validate these single cells as LGR5+ cells using another marker as well. It is already reported that there are regions in crypt where LGR5 cell population is high, medium and low. Author should use some other method to show the location of these single cells.

Thank you for your valuable comment.

We recognize that most studies on the intestinal epithelium have focused on the small intestine, with limited research on rectal stem cell dynamics, which is the focus of our work. Some of the representative markers of the small intestine are not useful for detecting rectal stem cells (ASCL2, OLFM4). In our study, we used LGR5 antibodies to stain cells morphologically identified as stem cells in the crypts of the rectal epithelium, as shown in Figure 1. These antibodies provide a stable marker for detecting Lgr5-positive cells, but we understand the need for further validation.

To address this, we are conducting a comprehensive RNA-seq analysis to identify the gene clusters that support Lgr5-positive cells in the rectum. This analysis will help us discover additional marker molecules that can be detected with greater specificity, allowing for more accurate identification and localization of Lgr5-positive cells.

3. In the method author mentioned that “The epithelial mucosa was carefully collected by soaking and freezing with TRIzol reagent (Invitrogen) and by scraping with a surgical blade”. Author should isolate the epithelial cells instead of mucosa, as mucosa contains all other stromal cells. So, all the qPCR analysis should be done on the Intestinal epithelial cells.

Thank you for your valuable comment.

In practice, we scrape off approximately one-third of the frozen mucosal surface, which we consider to be primarily epithelial tissue. We have revised the Materials and Methods section to clarify this procedure. The changes are added in section 4.5 (lines 361-363).

4. The IF image quality is very poor (Fig1, 3, 5)

Thank you for your valuable comment.

We have replaced the images with newly acquired high-resolution confocal microscopy images.

5. The experiments should be repeated using the clinical dose of fractionated radiation instead of single radiation dose to create the scenario of clinical proctitis as it develops in patients following radiation therapy.

Thank you for your valuable comment.

We initially conducted single irradiation experiments to establish baseline data on the radiosensitivity of the rectal epithelium. In our system, the mouse LD100 dose for single irradiation is 29.5-30 Gy. However, when calculating the biological effective dose (BED) for a four-fraction irradiation with an α/β value of 8 Gy, the derived single dose is 11.5 Gy, resulting in a total of 46 Gy across four fractions, which is an excessively high dose.

The single dose required to reach LD100 in a four-fraction regimen was significantly lower than 11.5 Gy. This suggests that the actual α/β values in the rectum are likely much higher than 8 Gy. We are currently conducting validation experiments to accurately determine the α/β values in the mouse rectum.

Given the complexity of determining appropriate fractionation conditions, we recognize the importance of thoroughly examining these parameters before setting split irradiation protocols. This will be the focus of our upcoming research efforts.

Reviewer 2 Report

Comments and Suggestions for Authors

The intestinal tract is highly radiosensitive, with RT-induced rectal injury being a critical side effect that limits RT doses in the abdominal and pelvic regions.  In this study, Lgr5-EGFP knock-in mice and LGR5 antibody staining in wild-type mice were utilized. After irradiation, qPCR analysis revealed a significant decrease in Lgr5 expression in the rectal mucosa. Stem cell population patterns varied with dose: at lower doses, Lgr5 clusters were observed, while at higher doses, scattered Lgr5-dotted populations emerged, with re-clustering aiding recovery. The threshold dose for disrupting these structures was 18 Gy. I think some critical points need to be addressed.
Please include the following details in the materials and methods section:

  • The duration of the experiment.

  • The specific criteria (humane endpoints) used to determine when euthanasia was necessary.

  • The frequency of monitoring animal health and behavior.

  • The number of animals used, euthanized, and any found dead, along with the cause of death for all animals.

  • The procedure used to confirm death in the study.

  • All animal welfare considerations, such as efforts to minimize suffering and distress, including the use of analgesics, anesthetics, or special housing conditions.

  • The route of administration and dosage of anesthetics used (typically in mg/kg for injectable anesthetics or % for inhalants).

  • The method of euthanasia employed.

  • Authors are encouraged to follow ARRIVE checklist (https://www.nc3rs.org.uk/arrive-guidelines)

Author Response

We have added all the necessary information in the Materials and Method section.

1. The duration of the experiment.

     Thank you for your valuable comment.

The total duration of this experiment was between May 2022 and August 2024 (Section 4.1, lines 308-309)

2. The specific criteria (humane endpoints) were used to determine when euthanasia was necessary.

      Thank you for your valuable comment.

The specific criteria was a 20 % decrease in body weight over 7 days (Section 4.1, lines 301-302).

3. The frequency of monitoring animal health and behaviour.

      Thank you for your valuable comment.

The animal health and behaviour were monitored diligently once every day throughout the study (Section 4.1, line 300).

4. The number of animals used, euthanized, and any found dead, along with the cause of death for all animals.

      Thank you for your valuable comment.

In total, approximately 88 Lgr5-EGFP mice and 130 ICR wild-type mice were utilized in this study. The exact number of animals per group is provided in the materials and method sections with group sizes (n) ranging from 3 to 10 (Section 4.1, lines 302-306). Except in Figures 4B and 4C, the number of mice at certain time points was reduced to n=2 due to high-dose mortality, statistical analysis could not be performed (Section 4.7, lines 391-394) For details, the minimal data is provided.

5.  The procedure used to confirm death in the study.

Thank you for your valuable comment.

The confirmation of death was performed through careful observation of the mouse carcasses (Section 4.1 line 301)

6. All animal welfare considerations, such as efforts to minimize suffering and distress, including the use of analgesics, anesthetics, or special housing conditions.

Thank you for your valuable comment.

Anesthetic reagent was used to accurately irradiate the caudal half-body and to reduce immobilization stress during the irradiation treatment. (Section 4.2 , lines 314-316)

7.  The route of administration and dosage of anesthetics used (typically in mg/kg for injectable anesthetics or % for inhalants).

Thank you for your valuable comment

The anesthesia was administered intraperitoneally. It was carefully calculated and administered in accordance with established protocols to ensure the safety and well-being of the animals. (Section 4.2, lines 312-313)

The dosage of anesthetics is provided in (mg/kg) in materials and methods ( Section 4.2, lines 313-314).

8. The method of euthanasia employed.

             Thank you for your constructive comment.

 The method of euthanasia used was cervical dislocation (Section 4.3, line 325).

9. Authors are encouraged to follow ARRIVE checklist.

 Thank you for your valuable comment.

We have newly added new information after referring to ARRIVE checklist (Section  4.1, lines 306-308)
